# Activation of Calpain Contributes to Mechanical Ventilation-Induced Depression of Protein Synthesis in Diaphragm Muscle

**DOI:** 10.3390/cells11061028

**Published:** 2022-03-18

**Authors:** Hayden W. Hyatt, Mustafa Ozdemir, Matthew P. Bomkamp, Scott K. Powers

**Affiliations:** 1Department of Applied Physiology and Kinesiology, University of Florida, Gainesville, FL 32611, USA; hhyatt3@jh.edu (H.W.H.); ozdemirm@ufl.edu (M.O.); mbomkamp@ufl.edu (M.P.B.); 2Department of Physiology, The Johns Hopkins University School of Medicine, Baltimore, MD 21201, USA; 3Department of Health Sciences, Stetson University, Deland, FL 32720, USA

**Keywords:** protein translation, oxidative stress, calpastatin, redox signaling, anabolic signaling, proteolysis

## Abstract

Mechanical ventilation (MV) is a clinical tool that provides respiratory support to patients unable to maintain adequate alveolar ventilation on their own. Although MV is often a life-saving intervention in critically ill patients, an undesired side-effect of prolonged MV is the rapid occurrence of diaphragmatic atrophy due to accelerated proteolysis and depressed protein synthesis. Investigations into the mechanism(s) responsible for MV-induced diaphragmatic atrophy reveal that activation of the calcium-activated protease, calpain, plays a key role in accelerating proteolysis in diaphragm muscle fibers. Moreover, active calpain has been reported to block signaling events that promote protein synthesis (i.e., inhibition of mammalian target of rapamycin (mTOR) activation). While this finding suggests that active calpain can depress muscle protein synthesis, this postulate has not been experimentally verified. Therefore, we tested the hypothesis that active calpain plays a key role in the MV-induced depression of both anabolic signaling events and protein synthesis in the diaphragm muscle. MV-induced activation of calpain in diaphragm muscle fibers was prevented by transgene overexpression of calpastatin, an endogenous inhibitor of calpain. Our findings indicate that overexpression of calpastatin averts MV-induced activation of calpain in diaphragm fibers and rescues the MV-induced depression of protein synthesis in the diaphragm muscle. Surprisingly, deterrence of calpain activation did not impede the MV-induced inhibition of key anabolic signaling events including mTOR activation. However, blockade of calpain activation prevented the calpain-induced cleavage of glutaminyl-tRNA synthetase in diaphragm fibers; this finding is potentially important because aminoacyl-tRNA synthetases play a central role in protein synthesis. Regardless of the mechanism(s) responsible for calpain’s depression of protein synthesis, these results provide the first evidence that active calpain plays an important role in promoting the MV-induced depression of protein synthesis within diaphragm fibers.

## 1. Introduction

Mechanical ventilation (MV) is a clinical tool used to maintain adequate pulmonary gas exchange in patients that require respiratory support during critical illness or surgery. Although MV is frequently a life-saving clinical tool, prolonged MV promotes inspiratory muscle weakness due to diaphragmatic atrophy and contractile dysfunction; collectively, this syndrome is labeled “ventilator-induced diaphragm dysfunction (VIDD)” [1]. VIDD is an important clinical problem because diaphragm dysfunction is likely a key contributor to problems in weaning patients from ventilators [2,3,4]. Complications in weaning are pervasive as ~40–70% of MV patients fail to wean during the first weaning effort [5,6]. The inability to wean patients from the ventilator extends hospitalization and is associated with both increased morbidity (e.g., infections) and mortality [5,7]. Currently, a detailed understanding of the mechanism(s) that promote VIDD does not exist, and thus, no standard clinical treatment exists. Therefore, improving our understanding of the cell-signaling events that contribute to VIDD is essential to identify molecular targets for clinical treatments.

Although numerous unanswered questions remain about the pathogenesis of VIDD, research reveals that VIDD results from both accelerated proteolysis and depressed protein synthesis [8]. In reference to proteolysis, it is established that prolonged MV activates all four major proteolytic systems (i.e., ubiquitin-proteasome, autophagy, caspase-3, and calpain) within diaphragm fibers [9,10,11,12,13]. To date, the role that calpains play in MV-induced proteolysis has received limited attention. Nonetheless, growing evidence indicates that active calpains play a key role in MV-induced proteolysis [10,14,15]. Furthermore, active calpains have been shown to inhibit the anabolic signaling of both Akt and the mammalian target of rapamycin (mTOR) in skeletal muscle [16]. This observation suggests that calpains could play a dual role in muscle wasting by both accelerating proteolysis and depressing protein synthesis [16,17]. This is a testable hypothesis and forms the foundation for the current experiments.

To investigate the role that active calpains play in the depression of protein synthesis in the diaphragm during prolonged MV, we used a well-established animal model of MV and transfected diaphragm muscle with a recombinant adeno-associated viral vector (i.e., rAAV9) to overexpress the calpastatin (CAST) transgene; calpastatin is a selective endogenous inhibitor of calpain activity [18]. Based on evidence that calpain depresses anabolic signaling in skeletal muscle, we hypothesized active calpain depresses both anabolic signaling and protein synthesis in diaphragm fibers during prolonged MV.

## 2. Materials and Methods

### 2.1. Animals and Institutional Approval

Young adult (~4–6 months old) female Sprague-Dawley rats were used in these experiments. Animals were housed in an AAALAC accredited vivarium where the room temperature was carefully maintained between 20 and 22 °C. Animals were maintained on a 12:12h light cycle with food and water provided ad libitum. After purchase from an accredited vendor, animals were allowed to acclimate in the animal housing facility for 7 days prior to experimentation. Animals were cared for in accordance with the Guide for the Care and Use of Laboratory Animals. These experiments were approved by the Institutional Animal Care and Use Committee at the University of Florida.

### 2.2. Experimental Design

To determine if active calpain promotes MV-induced depression of protein synthesis in the diaphragm, animals were randomly assigned to one of four experimental groups: (1) control (i.e., wildtype, spontaneously breathing animals; CON, n = 12); (2) wildtype animals exposed to 12 h of MV (MV, n = 11); (3) spontaneously breathing animals transfected with the CAST transgene (CON-CAST, n = 12); and (4) animals transfected with the CAST transgene and exposed to 12 h of MV (MV-CAST, n = 12).

### 2.3. Experimental Protocol

#### 2.3.1. Surgical Protocol for AAV-CAST Administration

To determine the effects of active calpain on diaphragm function, an AAV vector containing CAST was delivered directly to the diaphragm via intramuscular injection; complete details of this procedure can be found in Smuder et al. [19], and therefore, only a short summary is provided here. After reaching a surgical plane of anesthesia, a laparotomy was performed, followed by injections into the costal diaphragm with either sterile saline (CON and MV) or AAV-9-CAST (CON-CAST and MV-CAST). This technique for gene transfer effectively delivers the gene of interest to diaphragm muscle fibers without adverse side-effects [19]. The AAV vector containing CAST was obtained from Vector Biolabs (Malvern, PA, USA, #AAV-204134); gene expression was controlled by a CMV promoter sequence. Following surgery, animals were provided Buprenorphrine SR-LAB (1.2 mg/kg, Zoopharm, Swedesboro, NJ, USA) prior to awakening and every 12 h for 72 h during recovery. Prolonged MV experiments were performed four weeks following the surgical procedures.

#### 2.3.2. Acutely Anesthetized Control Animals

Animals assigned to control groups were anesthetized using IP injections of sodium pentobarbital (60 mg/kg body weight). After reaching a surgical plane of anesthesia, the costal diaphragm was removed, frozen in liquid nitrogen, and stored at −80 °C for subsequent analysis. Note that animals in the control groups were sacrificed at the same time-point as the animals assigned to the MV groups.

#### 2.3.3. Mechanical Ventilation

MV animals were exposed to 12 h of controlled MV. Briefly, animals were anesthetized with an IP injection of sodium pentobarbital (60 mg/kg body weight); after reaching a surgical plane of anesthesia, a tracheostomy was performed, and animals were connected to a pressure-controlled ventilator (Servo Ventilator 300; Siemens, Munich, Germany). Ventilator settings included a respiratory rate of 80/breaths per minute with the positive end-expiratory pressure established at 1cmH_2_O. The carotid artery was cannulated to permit measurement of blood pressure, and the jugular vein was cannulated to allow continuous administration of anesthesia (i.e., sodium pentobarbital/10 mg/kg/h). Arterial blood gasses were routinely measured using a blood gas analyzer (GEM Premier 3000; Instrumentation Laboratory, Lexington, MA, USA). If required to maintain blood gas homeostasis, adjustments to the ventilator settings were performed to control alveolar ventilation and adjust the fraction of inspired oxygen. Throughout prolonged MV, the PaO_2_ was maintained > 60 mmHg, and PaCO_2_ was sustained below 40 mmHg. Body temperature was controlled at ~37 °C by a heating blanket, and routine care to the experimental animal was provided to lubricate eyes, express the bladder, and remove airway mucus during the 12-h period of MV. To reduce airway secretions during MV, glycopyrrolate (0.02 mg/kg) was administered intramuscularly every two hours. Upon completion of 12 h of MV, the costal diaphragm was quickly removed, frozen in liquid nitrogen, and stored at −80 °C for subsequent analyses.

#### 2.3.4. Western Blot

Western blots were performed to determine the abundance of calpastatin, the calpain-specific cleaved fragment of αII-spectrin, puromycin, phosphorylated Akt, phosphorylated mTOR, phosphorylated p70 S6 kinase (p70S6K), phosphorylated eukaryotic initiation factor 4E binding protein 1 (4E-BP1), and cleaved glutaminyl-tRNA synthetase. Briefly, ~30 mg of costal diaphragm was homogenized in homogenization buffer (5 mM Tris-HCL, 5 mM EDTA; pH = 7.4) with a protease inhibitor cocktail (Sigma-Aldrich, St. Louis, MO, USA) and centrifuged at 1500× *g* for 10 min at 4 °C. Diaphragmatic supernatant was collected, and protein concentration was quantified (Bradford, Sigma-Aldrich). Proteins were then added to 2× Laemmli sample buffer (1610737, Bio Rad, Hercules, CA, USA) with 5% (*w*/*v*) dithiothreitol and boiled at 100 °C for 5 min. The proteins within the supernatant were then separated via polyacrylamide gel electrophoresis and transferred to a polyvinylidene difluoride membrane. Membranes were probed for CAST (Abnova, Taipei, Taiwan, #H00000831-B01), αII-spectrin (Santa Cruz, Dallas, TX, USA, sc-48382), puromycin (Kerafast, Boston, MA, USA # EQ0001), phosphorylated Akt (Ser 473 Cell Signaling Technology, Danvers, MA, USA #9271), total Akt (Cell Signaling Technology, #2938), phosphorylated mTOR (Cell Signaling Technology, #5536), total mTOR (Cell Signaling Technology, #2983), phosphorylated p70S6K (Cell Signaling Technology, #9205), total p70S6K (Cell Signaling Technology, #9202) phosphorylated 4E-BP1 (Cell Signaling Technology, #2855), total 4E-BP1 (Cell Signaling Technology, #9644), and cleaved glutaminyl-tRNA synthetase (Santa Cruz, Dallas, TX, USA sc-271078). Revert total protein stain (LI-COR Biosciences, Lincoln, NE, USA) was used to normalize for loading control. Membranes blotted for phosphorylated proteins were subsequently stripped using LI-COR NewBlot stripping buffer (LI-COR Biosciences) and probed for the respective total target protein. Membranes were imaged fluorescently and analyzed using the LI-COR Odyssey CLx Imaging System (LI-COR Biosciences).

#### 2.3.5. Measurement of In Vivo Protein Synthesis via the SUnSET Technique

The Surface Sensing of Translation (SUnSET) technique uses an anti-puromycin antibody for detection of puromycin-labeled peptides. This technique has been shown to be valid and reliable for the measurement of muscle protein synthesis in vivo (reviewed in Goodman and Hornberger) [20]. Briefly, animals received an IP injection of puromycin (InvivoGen, San Diego, CA, USA, ant-pr) 30 min prior to the experimental end-point. Specifically, each animal received 0.04 µMol/g body weight. Incorporation of puromycin into growing polypeptide chains (i.e., protein synthesis rate) was determined by immunoblotting for puromycin in total protein extracts via western blotting.

### 2.4. Statistical Analysis

Groups comparisons for all dependent variables were made by one-way analysis of variance (ANOVA), and when appropriate, a Tukey’s HSD (honestly significant difference) test was performed post hoc. Significance was established at *p* < 0.05. Data are expressed as mean ± SD. Statistical analyses were performed using Prism 9 (GraphPad Software, San Diego, CA, USA).

## 3. Results

### 3.1. Calpastatin Overexpression Prevents MV-Induced Calpain Activation in Diaphragm Muscle

To evaluate the level of transgene overexpression of CAST in diaphragm muscle, we measured the protein abundance of the overexpressed CAST construct in diaphragm fibers via Western blotting. Compared to CON, the abundance of CAST in diaphragm fibers was ~50–60% greater in CON-CAST animals (Figure 1A). Similarly, compared to MV, the CAST levels in the diaphragm of MV-CAST animals were significantly higher (i.e., ~70*–*80%) (Figure 1A). Together, these results confirm that transgene overexpression of CAST was effective in significantly increasing the abundance of CAST in the diaphragm of transfected animals.

To establish if CAST overexpression prevented the MV-induced activation of calpain in diaphragm muscle fibers, we determined the abundance of the 145 kDa calpain-specific αII-spectrin cleavage fragment as a biomarker of calpain activity. Notably, this 145 kDa spectrin cleavage fragment is widely accepted biomarker of calpain activation in vivo [21]. Our findings reveal that overexpression of CAST impeded the MV-induced calpain activation in diaphragm muscle fibers as indicated by the elevated abundance of the 145 kDa αII-spectrin cleavage product in diaphragm fibers of MV animals compared to all other experimental groups (i.e., CON, CON-CAST, and MV-CAST) (Figure 1B). Collectively, these data confirm that CAST was successfully overexpressed in the diaphragm of CON-CAST and MV-CAST animals and that the overexpression of CAST in diaphragm fibers averted the MV-induced increase in calpain activation.

### 3.2. Prevention of Calpain Activation in the Diaphragm Averted the MV-Induced Depression of Protein Synthesis

To determine if prevention of calpain activation protects against MV-induced decrease in protein synthesis, we measured in vivo protein synthesis in the diaphragm during the final 30 min of MV. As expected, compared to CON, 12 h of MV resulted in a significant decrease (~30–40%) in total protein synthesis in diaphragm muscle fibers (Figure 2). Notably, the transgene overexpression of CAST prevented MV-induced calpain activation and rescued the MV-induced decrease in protein synthesis in diaphragm muscle fibers. Importantly, overexpression of CAST in the diaphragm of spontaneously breathing animals (i.e., CON-CAST) did not significantly impact protein synthesis in the diaphragm; this observation confirms that the overexpression of CAST does not influence protein synthesis in the diaphragm of spontaneously breathing animals (Figure 2).

### 3.3. Prevention of MV-Induced Calpain Activation in the Diaphragm Does Not Prevent the Down-Regulation of Anabolic Signaling

Guided by published research, we hypothesized that MV-induced activation of calpain would result in a decrease in anabolic signaling. To test this prediction, we measured four important biomarkers of anabolic signaling. First, we determined the levels of both phosphorylated (i.e., activated) Akt and mTOR. Akt is an upstream stimulator of mTOR and active mTOR phosphorylates both p70S6K and 4E-BP1 to promote translation and increase protein synthesis [22]. Our results reveal that compared to CON and CON-CAST, prolonged MV resulted in a decrease in both phosphorylated Akt and mTOR in diaphragm fibers (Figure 3). In contrast to our hypothesis, overexpression of CAST and prevention of calpain activation did not protect against MV-induced decreases in the activation of both Akt and mTOR in diaphragm fibers.

We next determined the activation of two regulators of protein synthesis that are downstream from Akt and mTOR. Specifically, we measured the levels of phosphorylated p70S6 kinase and 4E-BP1 in the diaphragm. p70S6 kinase phosphorylates the S6 ribosomal protein to induce protein translation at the ribosome [22]. In contrast, 4E-BP1 is a translation repressor protein; however, phosphorylation of 4E-BP1 by mTOR prevents 4E-BP1 from interfering with the initiation of translation, thus promoting protein synthesis [22]. As expected, compared to both CON and CON-CAST, prolonged MV resulted in decreased levels of activated p70S6K and 4E-BP1 in the diaphragm. Contrary to our thesis, overexpression of CAST and prevention of calpain activation did not protect against MV-induced decreases in the phosphorylation of p70S6K and 4E-BP1 in diaphragm fibers (Figure 3).

We continued our search for the mechanistic connection between calpain activation and depressed protein synthesis by investigating diaphragm levels of the calpain-specific cleaved fragment of glutaminyl-tRNA synthetase; glutaminyl-tRNA synthetase is a member of the aminoacyl-tRNA synthetase family [23]. Aminoacyl-tRNA synthetases (ARSs) are a family of 20 essential enzymes that ligate amino acids to their corresponding tRNAs in protein synthesis; notably, functional ARSs are required for protein synthesis [24]. It follows that calpain cleavage of one or more aminoacyl-tRNA synthetase can negatively impact protein synthesis. Our results reveal that compared to CON, CON-CAST, and MV-CAST, diaphragmatic levels of the calpain-specific fragment of glutaminyl-tRNA synthetase were significantly elevated in diaphragm fibers during prolonged MV. In contrast, prevention of the MV-induced activation of calpain (via over-expression of CAST) rescued the calpain-mediated cleavage of glutaminyl-tRNA synthetase in diaphragm fibers (Figure 4).

## 4. Discussion

### 4.1. Summary of Experimental Results

These experiments provide new and important insight into the role that calpain activation plays in the regulation of protein synthesis in diaphragm muscle during prolonged MV. Specifically, we tested the hypothesis that active calpain depresses both anabolic signaling and protein synthesis in diaphragm fibers during prolonged MV. Our results support the hypothesis that calpain activation depresses protein synthesis in the diaphragm during prolonged MV; nonetheless, our findings do not support the position that this depression in protein synthesis occurs due to a calpain-mediated decrease in active Akt or mTOR. However, our findings suggest that prevention of MV-induced calpain activation rescues the depression in protein synthesis in the diaphragm by preventing calpain-mediated cleavage of glutaminyl-tRNA synthetase. A discussion of these results and a brief critique of our experimental approach follows.

### 4.2. Critique of Experimental Approach

To determine if calpain plays a required role in the MV-induced depression of protein synthesis in the diaphragm, we used a well-established preclinical model of VIDD. The rat was selected as the experimental model because rat and human diaphragm muscles share similar anatomical features, functional characteristics, and fiber type composition and exhibit a parallel time course in the development of VIDD [8,9,25]. Moreover, female rats were studied because they maintain a relatively constant body weight from 4 to 6 months of age, and notably, no gender differences exist in MV-induced diaphragmatic atrophy between female and male rats [25,26].

Numerous pharmacological inhibitors of calpain exist; however, most of these compounds lack specificity and exhibit off-target effects [27]. Hence, to avoid this experimental pitfall, we inhibited calpain activation via AAV-9 transgene overexpression of CAST in diaphragm fibers. Importantly, we have shown that this approach can overexpress target genes without adverse effects on muscle fibers [19]. We selected CAST to inhibit calpain activation because CAST is an endogenously expressed protein and the only known function of CAST is the inhibition of calpains [28]. Moreover, our prior experiments indicate that overexpression of CAST in the diaphragm does not influence the baseline size or contractile function of diaphragm muscle fibers [14].

### 4.3. Calpains Play an Essential Role in MV-Induced Depression of Protein Synthesis in Diaphragm Muscle

Again, our findings support the thesis that calpain activation depresses protein synthesis in the diaphragm during prolonged MV but do not support our hypothesis that this depression in protein synthesis occurs due to a calpain-mediated decrease in phosphorylated Akt or mTOR signaling. Therefore, to discern other potential mechanisms linking active calpain with depressed protein synthesis, we investigated calpain substrates that contribute to protein synthesis. This search led to the discovery that active calpain cleaves components within the ARS complex, and this cleavage decreases the functionality of these molecules. This is significant to protein synthesis because the first essential step of protein translation involves the covalent attachment of an amino acid to its cognate transfer RNA (tRNA). As discussed earlier, this procedure is performed by a highly specialized group of enzymes, the ARSs [29]. Notably, one ARS enzyme is designated for each amino acid. We focused on a specific ARS (glutaminyl-tRNA synthetase) because the calpain cleavage sites of this ARS are well-characterized [23]. Importantly, our results reveal that active calpain cleaves glutaminyl-tRNA synthetase in diaphragm muscle fibers during prolonged MV; it follows that a decline in functional glutaminyl-tRNA synthetase in the cell could limit protein synthesis. Hence, our discovery of calpain cleavage of glutaminyl-tRNA synthetase is a potential mechanism to explain a signaling link between active calpain and the depressed protein synthesis that occurs in the diaphragm during prolonged MV. This is a testable hypothesis that is worthy of future study. Nonetheless, the mechanisms responsible for MV-induced repression of protein synthesis are likely multi-factorial, and other calpain cleavage targets likely play a role.

Our hypothesis that active calpain depresses protein synthesis in diaphragm muscle during prolonged MV materialized from published results that active calpain reduces active Akt and mTOR in diaphragm muscle fibers; moreover, this report hypothesized that active calpain depresses muscle protein synthesis via inhibition of anabolic signaling [16]. While our findings support the hypothesis that active calpain depresses protein synthesis in diaphragm muscle, our data do not support the concept that the calpain-mediated depression of muscle protein synthesis occurs due to inhibition of Akt/mTOR signaling. Indeed, while overexpression of the CAST transgene prevented the MV-induced activation of calpain in the diaphragm, prevention of calpain activation did not rescue the depression in the levels of active Akt or mTOR. A definitive explanation for the divergent findings between the current work and the Smith and Dodd report is not clear; however, marked differences in the experimental techniques between the two studies provide clues. For example, the current study used a physiologically relevant in vivo model of MV-induced activation of calpain in the diaphragm muscle. In contrast, Smith and Dodd utilized an ex vivo experimental model whereby diaphragm fibers were incubated in a high calcium medium (3.5 mM) to activate calpain [16]. Previous work reveals that incubating skeletal muscle fibers in high calcium mediums not only activates calpain but also results in calcium-dependent muscle-wasting phenomena similar to certain diseases [30,31]. Moreover, in contrast to our approach of inhibiting calpain activation via endogenous overexpression of CAST, Smith and Dodd inhibited calpain via the pharmacological inhibitor calpeptin. Although calpeptin is an effective calpain inhibitor, off-target effects exist, including the inhibition of lysosomal proteases [27]. Therefore, the collective differences in the experimental approach between the current study and the work of Smith and Dodd provide a potential explanation for the divergent findings between the two investigations.

In a separate set of experiments, prior work by our laboratory has demonstrated that calpastatin overexpression prevents MV-induced oxidative stress. This work demonstrated that preventing calpain activation protects against MV-induced increases in mitochondrial emissions of reactive oxygen species (ROS) and increased 4-Hydroxynonenal (i.e., a well-established marker for lipid peroxidation) [14]. While our current work suggests that calpain cleavage of tRNA synthetases may contribute to the diminished protein synthesis rates with MV, it is also likely that inhibiting calpain activation may protect against diminished protein synthesis rates through decreased ROS emissions. In this regard, oxidative stress can play a direct role in decreasing rates of muscle protein synthesis [32,33]. Hence, MV-induced activation of calpains is likely to have direct (e.g., calpain-mediated cleavage of tRNA synthetases) and indirect (e.g., calpain-mediated elevation of mitochondrial ROS emissions) effects on protein synthesis rates in the diaphragm. Future studies are required to delineate the range of mechanisms responsible for calpain-mediated suppression of protein synthesis.

Which calpain isoforms are responsible for the MV-mediated depression of protein synthesis in the diaphragm? Calpain 1, calpain 2, and calpain 3 are all expressed in skeletal muscle [34]. However, overexpression of CAST does not inhibit calpain 3 activity [35]. Therefore, our finding that CAST overexpression protects the diaphragm against MV-induced decreases in protein synthesis indicates that calpain 1 and/or calpain 2 isoforms are the calpain isoforms responsible for this depression in protein synthesis in diaphragm fibers.

## 5. Summary and Conclusions

While MV can be a life-saving intervention for critically ill patients, an unwanted side-effect of prolonged MV is the diaphragmatic atrophy that occurs due to both accelerated proteolysis and depressed protein synthesis. This MV-induced diaphragmatic atrophy is significant because diaphragmatic atrophy leading to diaphragm weakness (i.e., VIDD) is a risk factor for difficulties in weaning patients from the ventilator [2,3,4]. At present, no standard clinical therapy exists to prevent VIDD. Obviously, a thorough understanding of the cell signaling events that promote diaphragmatic atrophy is a prerequisite to developing a pharmacological approach to avert MV-induced diaphragmatic wasting. In this regard, the current investigation supports the concept that calpains are a potential therapeutic target to combat VIDD. Indeed, our previous work indicates that active calpains play a key role in promoting accelerated proteolysis in the diaphragm during prolonged MV [14]. The current study markedly expands our previous results by demonstrating that prevention of MV-induced activation of calpain in the diaphragm protects against MV-induced decreases in muscle protein synthesis. Together, these important findings highlight the vital role that calpains play in the development of VIDD and provide insight into the mechanistic links between calpain and depressed protein synthesis in the diaphragm during prolonged MV.

## Figures and Tables

**Figure 1 cells-11-01028-f001:**
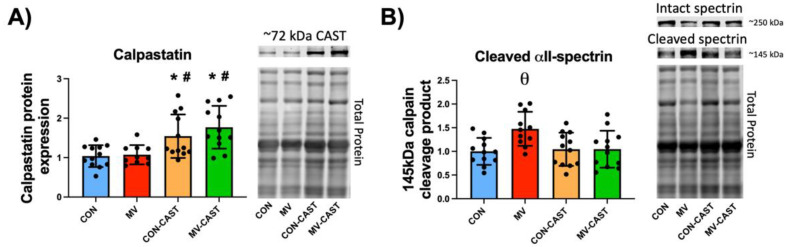
CAST overexpression prevents MV-induced activation of calpains. (**A**) Injection of AAV9-CAST into the diaphragm increase CAST expression in diaphragm fibers. (**B**) Overexpression of CAST in diaphragm fibers prevents MV-induced activation of calpains as determined by the calpain-specific cleavage fragment of αII-spectrin. * = significantly different from CON; # = significantly different from MV; θ = significantly different from CON, CON-CAST, and MV-CAST.

**Figure 2 cells-11-01028-f002:**
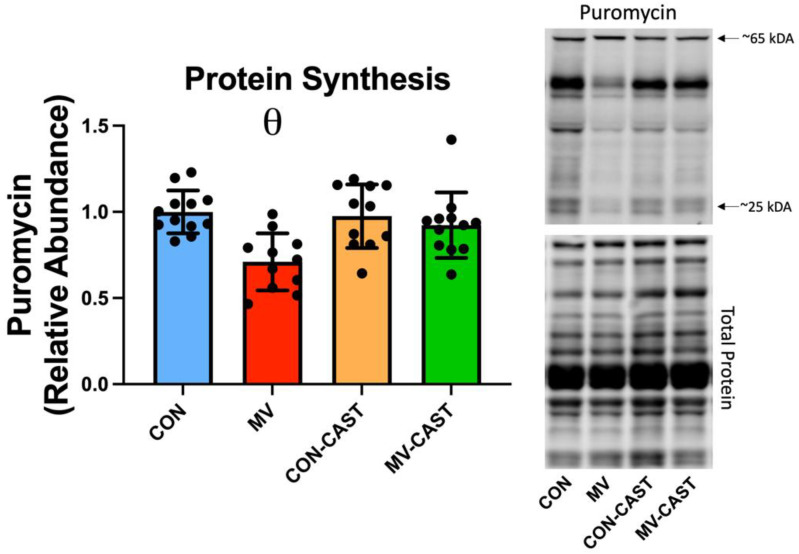
CAST overexpression protects the diaphragm against MV-induced decreases in protein synthesis. Relative rates of protein synthesis were determined by incorporation of the aminonucleoside puromycin into polypeptide chains. θ = MV is significantly different from CON, CON-CAST, and MV-CAST.

**Figure 3 cells-11-01028-f003:**
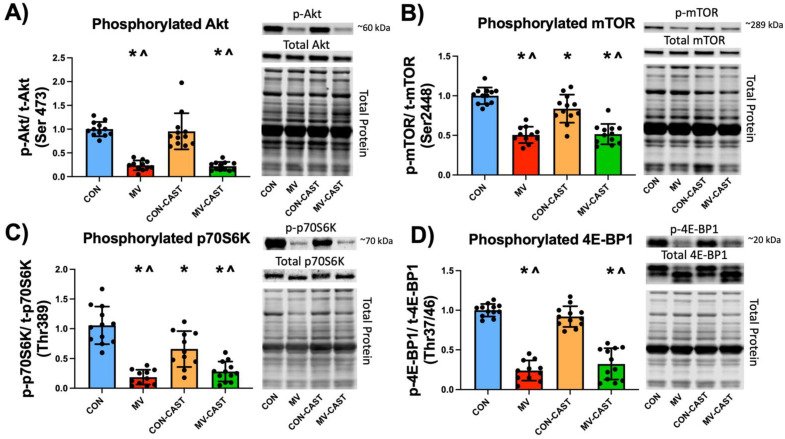
Markers of anabolic signaling via the Akt/mTOR pathway are decreased with MV. Notably, preventing MV-induced activation of calpains was not able to rescue diminished Akt/mTOR signaling. Protein expression of (**A**) Phosphorylated Akt (Ser473), (**B**) Phosphorylated mTOR (Ser2448), (**C**) Phosphorylated p70S6 kinase (Thr389), and (**D**) Phosphorylated 4E-BP1 (Thr37/46). * = significantly different from CON; ^ = significantly different from CON-CAST.

**Figure 4 cells-11-01028-f004:**
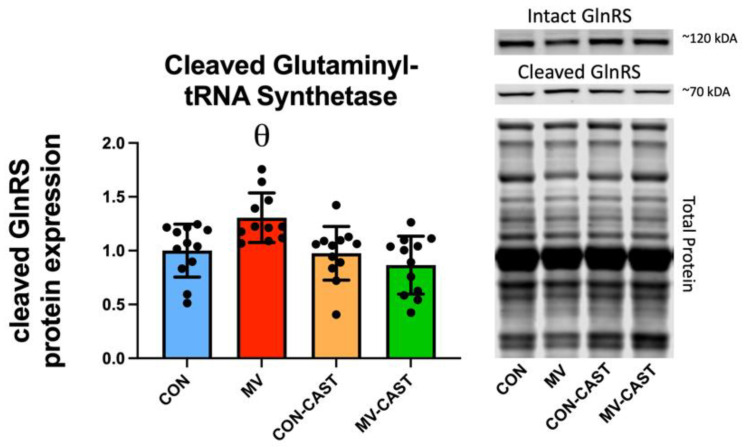
MV-induced activation of calpains results in increased cleavage of glutaminyl tRNA synthetase (GlnRS). θ = MV is significantly different from CON, CON-CAST, and MV-CAST.

## Data Availability

Individual data points are plotted within the figures contained in this publication.

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
