# Peer review of "Activation of Calpain Contributes to Mechanical Ventilation-Induced Depression of Protein Synthesis in Diaphragm Muscle"

_cells, 2022, doi:10.3390/cells11061028_

Round 1

Reviewer 1 Report

Using a rodent model of VIDD, Dr. Hyatt et al. try to determine if calpain plays a key role in the regulation of MV-induced depression of both anabolic signaling events and protein synthesis in diaphragm muscle. They found that blockade of calpain activation prevented the calpain-induced cleavage of glutaminyl-tRNA synthetase in diaphragm fibers. This study expands their previous results by demonstrating that prevention of MV-induced activation of calpain in the diaphragm protects against MV-induced decreases in muscle protein synthesis. However, phosphorylated Akt, mTOR, p70S6K, 4E-BP1 were not proved to regulating the anabolic signaling pathways in VIDD. There are some concerns that need to be addressed.

  1. Prolonged mechanical ventilation is defined as successful extubation after more than three spontaneous breathing trial or taking more than seven days. Can this rodent model of VIDD mimic clinical scenario of prolonged mechanical ventilation or just effects of short duration of mechanical ventilation?
  2. What are the effects of low-tidal-volume mechanical ventilation on muscle damage?
  3. Figure 1B: there is no differences between CON-CAST and MV-CAST. It means transgene overexpression of CAST can completely block the expression of α-II spectrin. Please explain the specificity of AAV-CAST. Is it clinical applicable?
  4. The data of Figure 1 have already been published by the authors in their 2021 Redox Biol paper. Why do authors add them in this manuscript?
  5. Which isoforms of Akt and mTOR are measured in this study?
  6. Why do authors choose intramuscular instead of intratracheal administration of AAC-CAST? Does this procedure induce complication of pneumothorax in animals?
  7. Do the authors have functional data, such as diaphragm contractile properties or thickening that AAV-CAST can rescue VIDD?
  8. How dose calpain regulates protein synthesis in VIDD? A discussion about other potential mechanisms are needed since the signaling pathways reviewed in previous publication (reference 7) is not proved in this study.

Author Response

Response to Reviewer 1

The authors thank this reviewer for their comments. Without question, your suggested changes have improved the quality of the revised manuscript. In the paragraphs below, we attempt to address each of the reviewer concerns in full.  

Reviewer comment: Prolonged mechanical ventilation is defined as successful extubation after more than three spontaneous breathing trial or taking more than seven days. Can this rodent model of VIDD mimic clinical scenario of prolonged mechanical ventilation or just effects of short duration of mechanical ventilation?

Author response:  Thank you for this comment.  To our knowledge, there is no “firm” definition of prolonged mechanical ventilation; however, 12-48 hours of mechanical ventilation (MV) is commonly labeled as prolonged in the clinical literature. Importantly, there is abundant evidence that preclinical studies of MV ranging from 12-48 hours closely mimics the human response to MV during the same time-period. Obviously, 12 hours of prolonged MV does not mimic the outcomes of 7 or more days of ventilatory support. See Petrof and Hussain (2016) and Jaber et al. (2011) for reviews on this topic.

Reviewer comment: What are the effects of low-tidal-volume mechanical ventilation on muscle damage?

Author response:  To our knowledge, this topic has only been addressed in one published report that concluded that changes in tidal volume (with or without PEEP) does not impact the magnitude of ventilator-induced diaphragm dysfunction. See Sassoon et al. Critical Care (2014) for details.

Reviewer comment: Figure 1B: there is no differences between CON-CAST and MV-CAST. It means transgene overexpression of CAST can completely block the expression of α-II spectrin. Please explain the specificity of AAV-CAST. Is it clinical applicable?

Author response: Thank you for this comment and question. The evidence provided in Figure 1B indicates that overexpression of a calpastatin in diaphragm fibers can prevent ventilator-induced activation of calpains and therefore, prevent the accumulation of the calpain-specific cleavage product (145kd) of alpha II spectrin. As stated in our manuscript, the only known effect of calpastain is the inhibition of calpain; therefore, calpastatin is highly specific in action. Lastly, given that prolonged MV activates calpain in the human diaphragm (Levine et al. NEJM, 2008), we believe that our findings have clinical relevance. Indeed, although transgene overexpression of calpastatin is not clinically applicable, the delivery of a pharmacological inhibitor of calpain is feasible in human patients.

Reviewer comment: The data of Figure 1 have already been published by the authors in their 2021 Redox Biol paper. Why do authors add them in this manuscript?

Author response: While we did report the same dependent measures (i.e., calpastatin and calpain cleaved alpha II spectrin) in our 2021 report, the current data are NOT from the same animals as the previously published study. Indeed, the data reported in Figure 1 are new measurements from the current experimental animals. We apologize for the confusion.

Reviewer comment: Which isoforms of Akt and mTOR are measured in this study?

Author response: Thank you for this question. The antibodies used in our study were directed at MTOR (which includes both the mTORC1 and mTORC2 complexes) and AKT1.

Reviewer comment: Why do authors choose intramuscular instead of intratracheal administration of AAC-CAST? Does this procedure induce complication of pneumothorax in animals?

Author response: Thank you for this question. The intramuscular injections into the diaphragm were conducted in order to overexpress calpastatin directly in the diaphragm in order to determine its role in diaphragm skeletal muscle. The injections are performed with 31g needles and do not create a large puncture. To our knowledge, our animals have never experienced a pneumothorax and are checked by veterinarian staff after each surgery. 

Reviewer comment: Do the authors have functional data, such as diaphragm contractile properties or thickening that AAV-CAST can rescue VIDD?

Author response: Thank you for this question. The answer is yes and these data were published in a previous study (see Hyatt et al. Redox Biology, 2021).

Reviewer comment: How dose calpain regulates protein synthesis in VIDD? A discussion about other potential mechanisms are needed since the signaling pathways reviewed in previous publication (reference 7) is not proved in this study.

Author response: Thank you for bringing this to our attention. We have briefly addressed this issue in the revised manuscript (lines 336-349). This segment highlights our previous finding that calpain inhibition protects against mitochondrial reactive oxygen species emission (ROS). ROS have been shown to affect rates of protein synthesis and this is also likely to play a role.

Reviewer 2 Report

   MV is a clinical tool that provides respiratory support to patients unable to maintain adequate alveolar ventilation but prolonged MV increases the risk of undesired side-effect diaphragmatic atrophy. The roles of active calpain in the MV-induced depression of both anabolic signaling events and protein synthesis in diaphragm muscle were investigated in this manuscript. It revealed that overexpression of calpastatin averted MV-induced activation of calpain in diaphragm fibers and rescued the MV-induced depression of protein synthesis in diaphragm muscle. The topic is interesting and the experimental designs were reasonable. It is worthy to be published after minor revision.

      1.The format of references list should follow the requirements of the Journal.  

  1. In the Section “Animals and Institutional Approval”, what criteria for caring the laboratory animals should be addressed.

Author Response

Response to reviewer 2

The authors thank this reviewer for their comments. We have addressed each of your concerns in full.  The following changes have been made in the revised manuscript.

Reviewer comment: The format of references list should follow the requirements of the Journal.  

Author response: Thank you for this reminder. The revised manuscript now complies with MDPI guidelines for authors.

Reviewer comment: In the Section “Animals and Institutional Approval”, what criteria for caring the laboratory animals should be addressed.

Author response: The revised manuscript provides additional information about care of laboratory animals (lines 70-75).

Reviewer 3 Report

This study was an extension of their recent published work (Redox Biology 2021). The finding that inhibition of calpain by calpastatin prevented MV-induced depressed protein synthesis in rat diaphragm is potentially interesting and the manuscript was well written and easy to read. However, data used to support the involvement of GlnRS-dependent mechanism is not convincing based on the following two facts. First, the representative western blot for cleaved GlnRS shows almost no change or very mild increase in MV compared with other groups; and second, it seems that intact GlnRS does not decrease in MV.  It would be nice if authors could show if calpain could affect multiple aminoacyl-tRNA synthetase complex in their system.

Author Response

REVIEWER 3

Reviewer comment: This study was an extension of their recent published work (Redox Biology 2021). The finding that inhibition of calpain by calpastatin prevented MV-induced depressed protein synthesis in rat diaphragm is potentially interesting and the manuscript was well written and easy to read. However, data used to support the involvement of GlnRS-dependent mechanism is not convincing based on the following two facts. First, the representative western blot for cleaved GlnRS shows almost no change or very mild increase in MV compared with other groups; and second, it seems that intact GlnRS does not decrease in MV.  It would be nice if authors could show if calpain could affect multiple aminoacyl-tRNA synthetase complex in their system.

RESPONSE TO REVEIWER 3:

Thank you for your thoughtful comments. These experiments were designed to test the hypothesis that active calpain plays a key role in the mechanical ventilation(MV)-induced depression of both anabolic signaling events and protein synthesis in diaphragm muscle. Our results clearly support the postulate that active calpain promotes a depression in diaphragmatic protein synthesis but do not support the prediction that active calpain depresses protein synthesis, in part, due to decreased anabolic signaling. Therefore, our experiments achieved the original goal.

After determining that active calpain does depress protein synthesis via changes in Akt, mTOR, and two down-stream regulators of protein synthesis, we performed additional experiments in an effort to determine the mechanism(s) responsible for our results. These experiments led to the possibility that calpain cleaves glutaminyl-tRNA synthetase (GlnRS). Analysis of our data reveals a ~17% decrease in intact GlnRS in the diaphragm following prolonged mechanical ventilation compared to control animals (3.9% increase in CON-CAST and 1.4% decrease in MV-CAST animals). These changes explain the ~30% increase in the ratio of cleaved GlnRS to totalGlnRS. In regard to the appearance of the gel, we made minor adjustments to the contrast in the representative image for the figure to better visualize the 17% decrease that exists for total GlnRS. While this is a modest change, it is feasible that this change can negatively impact rates of protein synthesis.

We agree with your comment that it will be ideal if our results were supported by further evidence of cleavage of additional tRNA synthetases. Our line of questioning regarding tRNA synthetases and calpain cleavage was sparked by findings of Lei et al (PMID: 26324710), which characterized several cleavage sites of tRNA synthetases by calpain. One of the most characterized cleavage sites by their work was that of GlnRS. The antibody they used recognized the epitope residues of 620-680 in GlnRS.

In this regard, we recognized that the Santa Cruz antibody offered a similar targeting region of amino acids 513-775 (sc-271078). This was the first tRNA synthetase target we assessed. Upon finding that there were increased cleavage fragments with MV and that calpastatin suppressed the appearance of these fragments, we set out to further characterize other cleavage fragments of additional tRNA synthetases. We also tried IleRS (sc-271826, targeted to aa 781-999), ArgRS (no information on epitope, sc-100990), LysRS (aa490-519, sc-393645). Unfortunately. We were unable to reliably detect cleavage fragments at the correct predicated molecular weights. Thus, it is plausible that these cleavage fragments occurred, but we were unable to detect them with the antibodies that we utilized.

In regard to the possibility of performing additional experiments, unfortunately, our experiments to date have consumed the entire ~500 mg of diaphragm tissue from our experimental animals. Nonetheless, our experiments have provided novel and important information about the impact of calpain activation on diaphragm muscle protein synthesis during prolonged mechanical ventilation. We have included additional statements for the need of further study on the mechanisms responsible for this calpain-protein synthesis interaction (lines 309-310 and 336-349).

Round 2

Reviewer 3 Report

No further comments